# Semi-supervised Vision Transformers at Scale

**Zhaowei Cai, Avinash Ravichandran, Paolo Favaro, Manchen Wang,
Davide Modolo, Rahul Bhotika, Zhuowen Tu, Stefano Soatto**

AWS AI Labs

{zhaoweic,ravinash,pffavaro,manchenw,dmodolo,ztu,soattos}@amazon.com

## Abstract

We study semi-supervised learning (SSL) for vision transformers (ViT), an under-explored topic despite the wide adoption of the ViT architecture to different tasks. To tackle this problem, we use a SSL pipeline, consisting of first *un/self-supervised pre-training*, followed by *supervised fine-tuning*, and finally *semi-supervised fine-tuning*. At the semi-supervised fine-tuning stage, we adopt an exponential moving average (EMA)-Teacher framework instead of the popular FixMatch, since the former is more stable and delivers higher accuracy for semi-supervised vision transformers. In addition, we propose a *probabilistic pseudo mixup* mechanism to interpolate unlabeled samples and their pseudo labels for improved regularization, which is important for training ViTs with weak inductive bias. Our proposed method, dubbed *Semi-ViT*, achieves comparable or better performance than the CNN counterparts in the semi-supervised classification setting. Semi-ViT also enjoys the scalability benefits of ViTs that can be readily scaled up to large-size models with increasing accuracy. For example, Semi-ViT-Huge achieves an impressive 80% top-1 accuracy on ImageNet using only 1% labels, which is comparable with Inception-v4 using 100% ImageNet labels. The code is available at https://github.com/amazon-science/semi-vit.

## 1 Introduction

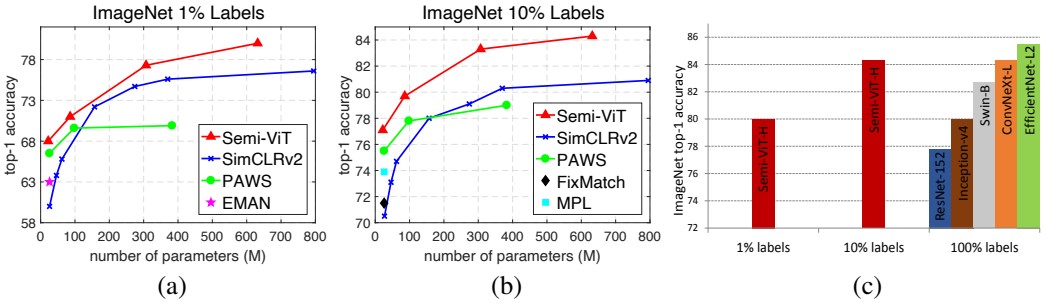

Figure 1: (a) and (b) are the comparisons of our Semi-ViT with the state-of-the-art SSL algorithms at different model scales, and (c) is the comparison with the state-of-the-art supervised models.

In the past few years, Vision Transformers (ViT) [18], which adapt the transformer architecture [64] to the visual domain, have achieved remarkable progress in supervised learning [63, 44, 73], un/self-supervised learning [16, 12, 25], and many other computer vision tasks [11, 19, 1, 58] (with architecture modifications). However, ViTs have yet to show the same advantage in semi-supervised learning (SSL), where only a small subset of the training data is labeled, a problem in the middle between supervised and un/self-supervised learning. Although several recent methods in SSL have

36th Conference on Neural Information Processing Systems (NeurIPS 2022).

significantly advanced the field [39, 62, 7, 55, 70, 10, 53], the transfer of these methods from Convolutional Neural Networks (CNN) to ViT architectures has yet to show much promise. For example, as discussed in [68], the direct application of FixMatch [55], one of the most popular SSL methods, to ViT leads to inferior performance (about 10 points worse) than when used with a CNN architecture. The challenge could be potentially caused by the fact that ViTs are known to require more data for training and to have a weaker inductive bias than CNNs [18]. However, in this paper we show that semi-supervised ViTs can outperform the CNN counterparts when trained properly, suggesting promising potential to advance SSL beyond CNN architectures.

To achieve that, we use the following SSL pipeline: 1) *un/self-supervised pre-training* on all data (both labeled and unlabeled), followed by 2) *supervised fine-tuning* only on labeled data, and finally 3) *semi-supervised fine-tuning* on all data. This pipeline is stable and helps reduce the sensitivity of hyperparameter tuning when training ViTs for SSL in our experiments. At the final stage of *semi-supervised fine-tuning*, we adopt the EMA-Teacher framework [62, 10], an improved version of the popular FixMatch [55]. Unlike FixMatch, which often fails to converge when training semi-supervised ViT, the EMA-Teacher shows more stable training behaviors and better performance. In addition, we propose *probabilistic pseudo mixup* for pseudo-labeling based SSL methods, a method that interpolates the unlabeled samples coupled with pseudo labels for enhanced regularization. In the standard mixup [75] the mixup ratio is randomly sampled from a Beta distribution. In contrast, in the *probabilistic pseudo mixup* the ratio depends on the respective confidence of two mixed-up samples, such that the sample with higher confidence will weigh more in the final interpolated sample. This new data augmentation technique brings non-negligible gains since ViT has weak inductive bias, especially for scenarios where the training is more difficult, *e.g.*, without un/self-supervised pre-training or on data regimes with very few labeled samples (*e.g.*, 1% labels). We call our method *Semi-ViT*. Notice that Semi-ViT is built on exactly the same design of ViTs (*i.e.*, there are neither additional parameters nor architectural changes).

Semi-ViT achieves promising results on several fronts (Figure 1). 1) For the first time, we show that *pure* ViTs can reach comparable or better accuracy than CNNs on SSL[1]. 2) Semi-ViT can be readily scaled up under the SSL setting. This is illustrated in Figure 1 (a) and (b) on ViT architectures at different scales, ranging from ViT-Small to ViT-Huge, and Semi-ViT outperforms the prior art such as SimCLRv2 [15]. 3) Semi-ViT has shown the potential for a substantial reduction of labeling cost. For example, as seen in Figure 1 (c), Semi-ViT-Huge with 1% (10%) ImageNet labels achieves comparable performance of a fully-supervised Inception-v4 [59] (ConvNeXt-L [45]). This implies a $100\times$ ($10\times$) reduction in human annotation cost. 4) Semi-ViT achieves the state-of-the-art SSL results on ImageNet, *e.g.*, 80.0% (84.3%) top-1 accuracy with only 1% (10%) labels. In addition, the substantial boost in performance by Semi-ViT is not isolated on ImageNet: we find an increase of 13%-21% (7%-10%) top-1 accuracy with 1% (10%) labels over the supervised fine-tuning baselines, for other datasets including Food-101 [9], iNaturalist [30] and GoogleLandmark [52].

## 2 Semi-supervised Vision Transformers

### 2.1 Pipeline

Some pipelines for semi-supervised learning exist in the literature. For example: 1) the model is directly trained from scratch using SSL techniques, *e.g.*, FixMatch [55]; 2) the model is un/self-supervised pretrained first and finetuned on labeled data later [26, 14, 22]; 3) the model is self-supervised pretrained first and then finetuned via semi-supervised learning on both labeled and unlabeled data [10]. In this paper, we instead adopt the following pipeline: first, optional *self-supervised pre-training* on all data without using any labels; next, standard *supervised fine-tuning* on available labeled data; and finally, *semi-supervised fine-tuning* on both labeled and unlabeled data. This procedure is similar to [15], with the difference that they use knowledge distillation [29] in their final stage. We find that this training pipeline trains semi-supervised vision transformers in a stable manner and achieves promising results, with possibly less hyperparameter tuning.

---

[1]Although [68] was the first to use a transformer architecture for SSL, it does so by combining both CNN and ViT architectures, and requires to use CNN as the teacher to produce pseudo labels.

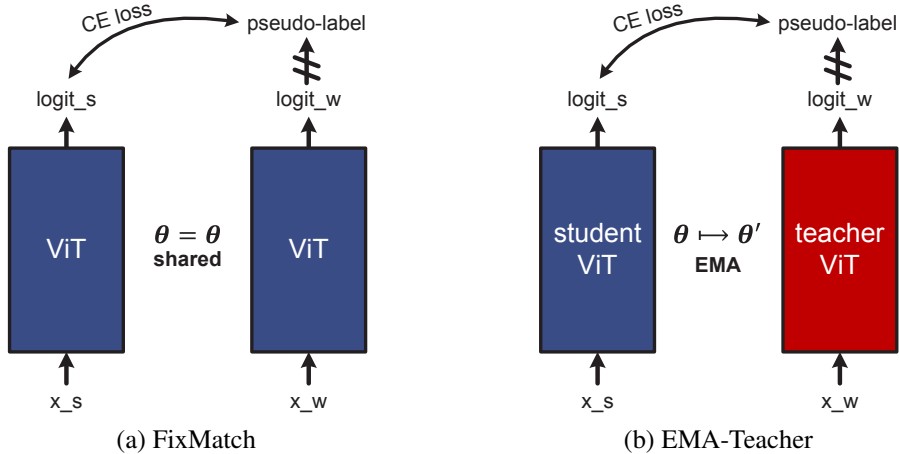

(a) FixMatch                  (b) EMA-Teacher

Figure 2: Comparison between FixMatch (a) and EMA-Teacher (b). $x_s/x_w$ is the strongly/weakly augmented view of a sample $x$, and $\theta$ are the model parameters.

## 2.2 EMA-Teacher Framework

FixMatch [55] emerged as a popular SSL method in the past few years. As discussed in [10], it can be interpreted as a student-teacher framework, where the student and teacher models are identical, as seen in Figure 2 (a). However, FixMatch has unexpected behaviors, especially when the model incorporates batch normalization (BN) [35]. Although ViT uses Layer Normalization (LN) [4] instead of BN as normalization, we still found that the FixMatch with ViT underperforms the CNN counterparts and often does not converge. This phenomenon was also observed in [68]. A potential reason to this is that the student and the teacher models are identical in FixMatch, which could easily lead to model collapse [26, 22]. This instability of the identical student-teacher framework has also been observed in other areas, e.g. semi-supervised speech recognition [42, 48, 28]. As suggested in [10], the EMA-Teacher (shown in Figure 2 (b)) is an improved version of the FixMatch, thus we adopt it for our Semi-ViT. In the EMA-Teacher framework, the teacher parameters $\theta'$ are updated by the exponential moving average (EMA) from the student parameters $\theta$,

$$\theta' := m\theta' + (1 - m)\theta, \tag{1}$$

where the momentum decay $m$ is a number close to 1, *e.g.*, 0.9999. The student parameters are updated by standard learning optimization, *e.g.*, SGD or AdamW [46]. The other components are exactly the same as the FixMatch, as seen in Figure 2. This temporal weight averaging can stabilize the training trajectories [3, 36] and avoids the model collapse issue [26, 22]. Our experiments also show this EMA-Teacher framework has better results and more stable training behaviors than FixMatch when training Semi-ViT.

## 2.3 Semi-supervised Learning Formulation

In the EMA-Teacher framework, there are both labeled and unlabeled samples in a minibatch during training. The loss on the labeled samples $\{(x_i^l, y_i^l)\}_{i=1}^{N_l}$ is the standard cross-entropy loss, $\mathcal{L}_l = \frac{1}{N_l} \sum_{i=1}^{N_l} CE(x_i^l, y_i^l)$. For an unlabeled sample $x^u \in \{x_i^u\}_{i=1}^{N_u}$, a weak and a strong augmentation are applied to it, generating $x^{u,w}$ and $x^{u,s}$, respectively. The weak augmented $x^{u,w}$ is forwarded through the teacher network, and outputs the probabilities over classes, $p = f(x^{u,w}; \theta')$. Then the pseudo label is produced by $\hat{y} = \arg\max_c p_c$ with its associated confidence $o = \max p_c$. The pseudo label with confidence higher than a confidence threshold $\tau$ is then used to supervise the learning of the student on the strong augmented sample $x^{u,s}$,

$$\mathcal{L}_u = \frac{1}{N_u} \sum_{i=1}^{N_u} [o_i \geq \tau] CE(x_i^{u,s}, \hat{y}_i), \tag{2}$$

where $[\cdot]$ is the indicator function. The overall loss is $\mathcal{L} = \mathcal{L}_l + \mu\mathcal{L}_u$, where $\mu$ is the trade-off weight. Note that only the pseudo labels with confidence higher than a threshold contribute to the final loss;

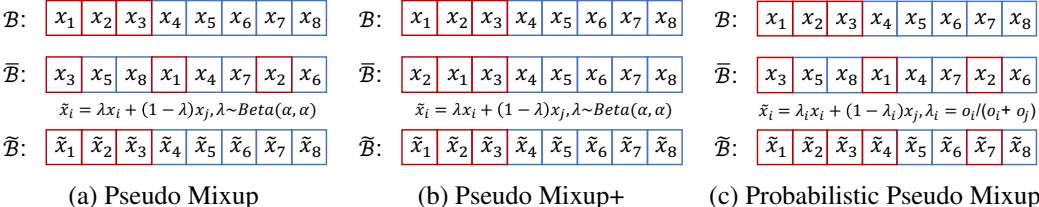

(a) Pseudo Mixup      (b) Pseudo Mixup+     (c) Probabilistic Pseudo Mixup

Figure 3: Different variations of mixup on unlabeled data. The red samples are the ones passing the confidence threshold, but not the blue samples.

the others are instead not used. The philosophy behind this filtering is that the pseudo labels with low confidence are noisier and could hijack the SSL training.

# 3 Probabilistic Pseudo Mixup

## 3.1 Mixup

Mixup [75] performs convex combinations of pairs of samples and their labels,

$$
\begin{aligned}
\tilde{x} &= \lambda x_i + (1 - \lambda)x_j, \\
\tilde{y} &= \lambda y_i + (1 - \lambda)y_j,
\end{aligned}
\tag{3}
$$

where the mixup ratio $\lambda \sim Beta(\alpha, \alpha) \in [0, 1]$, for $\alpha \in (0, \infty)$. The samples are mixed-up usually in a single minibatch during training. Given a minibatch $\mathcal{B}$ and its shuffled version $\bar{\mathcal{B}}$, the mixed-up minibatch is $\tilde{\mathcal{B}} = \lambda \mathcal{B} + (1 - \lambda)\bar{\mathcal{B}}$, where $\lambda$ could be either batch-wise or element-wise. Due to the nature of weak inductive bias, ViT is more data hungry than CNN, thus effective data augmentation, *e.g.*, mixup, is critical for training fully-supervised ViT [18, 63, 44, 73]. This also applies to Semi-ViT since it inherits the nature of weak inductive bias from ViT. Although it is standard to use mixup in supervised learning, how to employ it under pseudo-labeling based SSL framework, *e.g.*, the EMA-Teacher, is still unclear yet, and we are going to discuss it next.

## 3.2 Pseudo Mixup

Under the pseudo-labeling based SSL framework [40, 55, 53, 10], given an unlabeled sample and its pseudo label $(x^u, \hat{y})$, only when its confidence $o$ is not smaller than the confidence threshold $\tau$, it will contribute to the loss $\mathcal{L}_u$, as seen in (2). According to their confidence scores, the unlabeled minibatch $\mathcal{B}^u$ can be grouped into a clean subset $\hat{\mathcal{B}}^u = \{(x_i^u, \hat{y}_i)|o_i \geq \tau\}$ and a noisy subset $\dot{\mathcal{B}}^u = \mathcal{B}^u - \hat{\mathcal{B}}^u$. One straightforward solution is to apply mixup on the full unlabeled minibatch $\mathcal{B}^u$, with no differentiation between clean and noisy samples, denoted as *pseudo mixup*, as show in Figure 3 (a). After the pseudo mixup, still only the samples in $\hat{\mathcal{B}}^u$ contribute to the loss, and the samples in $\dot{\mathcal{B}}^u$ are abandoned. In this way, the mixup operation is more than just a data augmentation. In fact, a sample in $\dot{\mathcal{B}}^u$ will also contribute to the final loss if it is mixed-up with a sample in $\hat{\mathcal{B}}^u$. As a result, it could involve a substantial number of noisy samples into the loss calculation due to the randomness, which, however, is against the philosophy of pseudo-labeling. Since only the clean subset $\hat{\mathcal{B}}^u$ contributes to the final loss, another choice is to use mixup only on $\hat{\mathcal{B}}^u$, denoted as *pseudo mixup+*, as shown in Figure 3 (b). In this way, no sample in the noisy subset $\dot{\mathcal{B}}^u$ will affect the training.

## 3.3 Probabilistic Pseudo Mixup

Although the samples in $\dot{\mathcal{B}}^u$ are noisy, they still carry some useful information for the model to learn. The *pseudo mixup* above can somehow leverage those information by blending the noisy and clean pseudo samples together. However, the problem is the mixup ratio is randomly generated from a Beta distribution, which does not depend on the confidence of each sample. This is not ideal. For example, when two samples are mixed-up, the sample with higher confidence should have higher mixup ratio,

Table 1: Semi-ViT results comparing with fine-tuning. The models are self-pretrained by MAE [25].

| Model | Param | Method | 1% | 10% | 100% |
|---|---|---|---|---|---|
| ViT-Base | 86M | finetune | 57.4 | 73.7 | 83.7 |
| | | Semi-ViT | 71.0 | 79.7 | - |
| ViT-Large | 307M | finetune | 67.1 | 79.2 | 86.0 |
| | | Semi-ViT | 77.3 | 83.3 | - |
| ViT-Huge | 632M | finetune | 71.5 | 81.4 | 86.9 |
| | | Semi-ViT | 80.0 | 84.3 | - |

Table 2: The comparison between the FixMatch and the EMA-Teacher. ✘ means the training is failed with accuracy close to 0.

| Model | Pretrained | Method | 1% | 10% |
|---|---|---|---|---|
| ViT-Small | None | FixMatch | - | ✘ |
| | | EMA-Teacher | - | 65.6 |
| ViT-Base | None | FixMatch | - | ✘ |
| | | EMA-Teacher | - | 68.9 |
| ViT-Base | MAE | FixMatch | ✘ | 74.8 |
| | | EMA-Teacher | 65.3 | 78.1 |

such that it can weigh more in the final loss. Motivated by this intuition, we propose *probabilistic pseudo mixup* (Figure 3 (c)), where the mixup ratio $\lambda$ reflects the sample confidence,

$$\lambda_i = o_i/(o_i + o_j). \tag{4}$$

Also, the confidence score of $x_i^u$ is updated after the mixup operation as

$$o_i^* = \max(o_i, o_j), \tag{5}$$

because the confidence score should align with the majority of the image content. The final clean subset $\tilde{\mathcal{B}}^u = \{(\tilde{x}_i^u, \tilde{y}_i^u)|o_i^* \geq \tau\}$ will contribute to the final loss. *Probabilistic pseudo mixup* can enhance regularization, leverage information from all samples, even the noisy ones, and not violate the philosophy of pseudo labeling at the same time. It can effectively alleviate the issue of weak inductive bias of Semi-ViT and bring substantial gains, as will be shown in our experiments.

## 4 Experiments

We evaluate Semi-ViT mainly on ImageNet, which consists of ∼1.28M training and 50K validation images. We sample 10%/1% labels from the ImageNet training set for the semi-supervised evaluation. We study both scenarios: with and without self-supervised pre-training. Without self-pretraining, we only evaluate on 10% labels, since learning from scratch on 1% labels is very difficult. When self-pretrained, MAE [25] is mainly used, and we directly use their pretrained models. All learning is optimized with AdamW [46], using cosine learning rate schedule, with a weight decay of 0.05. The default momentum decay $m$ of (1) is 0.9999. In a minibatch, $N_u = 5N_l$, and the loss trade-off $\mu = 5$. The mixup is a combination of mixup [75] and Cutmix [74] as in the implementation of [69]. More details can be found in the appendix.

### 4.1 Semi-ViT Results

When the model is self-pretrained by MAE [25], we first evaluate the fine-tuning performance of MAE on the labeled data only, as the common practice in self/un-supervised learning literature [26, 14, 22], with results shown in Table 1. This already leads to strong semi-supervised baselines, *e.g.*, 81.4 top-1 accuracy for ViT-Huge on 10% labels, indicating that MAE is a strong self-supervised learning technique. However, Semi-ViT has additional significant improvements over the strong baselines for all models, *e.g.*, 8.5-13.6 points for 1% labels and 2.9-6.0 points for 10% labels. The fine-tuning results on 100% data are provided as upper-bounds for our Semi-ViT, and their gaps to Semi-ViT are small, *e.g.*, 4.0/2.7/2.6 points for ViT-Base/Large/Huge on 10% labels. An interesting observation is that the larger model is more effective for smaller number of labels, which is consistent with the observations in [15]. For example, the fine-tuning gaps between 1% and 100% labels are 26.3/18.9/15.4 points for ViT-Base/Large/Huge, which are decreasing. The observation on Semi-ViT

Table 3: The comparison among different mixup variations.

| Model | Pretrained | Mixup | 1% | 10% |
|---|---|---|---|---|
| ViT-Small | None | EMA-Teacher | - | 65.6 |
| | | Pseudo Mixup | - | 68.3 |
| | | Pseudo Mixup+ | - | 68.8 |
| | | ProbPseudo Mixup | - | 70.9 |
| ViT-Base | None | EMA-Teacher | - | 68.9 |
| | | Pseudo Mixup | - | 71.6 |
| | | Pseudo Mixup+ | - | 72.1 |
| | | ProbPseudo Mixup | - | 73.5 |
| ViT-Base | MAE | EMA-Teacher | 65.3 | 78.1 |
| | | Pseudo Mixup | 69.5 | 78.3 |
| | | Pseudo Mixup+ | 70.1 | 78.7 |
| | | ProbPseudo Mixup | 71.0 | 79.7 |

Table 4: The ablation on the confidence threshold.

| Method (ViT-Base) | label | $\tau = 0$ | $\tau = 0.3$ | $\tau = 0.4$ | $\tau = 0.5$ | $\tau = 0.6$ | $\tau = 0.7$ | $\tau = 0.8$ | $\tau = 0.9$ |
|---|---|---|---|---|---|---|---|---|---|
| EMA-Teacher | 1% | 63.1 | 64.4 | 64.6 | 65.1 | 65.3 | 65.1 | 64.4 | 63.4 |
| EMA-Teacher | 10% | 75.4 | 76.7 | 77.2 | 77.7 | 77.9 | 78.1 | 78.2 | 77.9 |
| Semi-ViT | 1% | 70.8 | 71.4 | 71.3 | 71.3 | 71.0 | 70.4 | 68.6 | 61.8 |
| Semi-ViT | 10% | 79.4 | 79.5 | 79.7 | 79.7 | 79.6 | 79.4 | 79.0 | 77.2 |

results is similar, *e.g.*, 12.7/8.7/6.9 points to their upper-bounds on 1% labels. These results have shown that vision transformers can also perform very well in semi-supervised learning, as well as supervised learning and un/self-supervised learning.

## 4.2 Ablation Studies

We ablate different factors of Semi-ViT in this section.

**FixMatch v.s. EMA-Teacher** is compared in Table 2. These experiments do not use the pseudo mixup techniques of Section 3 yet. When the model is not self-pretrained, the training of the FixMatch is unstable and often failed. When the model is self-pretrained, the FixMatch training becomes stable, and starts to achieve reasonable results, *e.g.*, 74.8 for ViT-Base on 10% labels, which is already better than the prior art on ResNet-50, *e.g.*, 73.9 of MPL [53] and 74.0 of EMAN [10]. But it is only 1.1 points higher than the fine-tuning baseline of Table 1, indicating the FixMatch is not an effective SSL framework for ViT. But the EMA-Teacher achieves much better results, 3.3 points of improvement over FixMatch when self-pretrained. Even without self-pretraining, the EMA-Teacher can still achieve satisfactory performance, while FixMatch fails.

**Probabilistic Pseudo Mixup** Different mixup variations on unlabeled data are compared in Table 3. Note that the standard mixup with the implementation of [69] is used on the labeled data as usual. The EMA-Teacher does not use any mixup mechanism on the unlabeled data, serving as baselines here. When *pseudo mixup* of Figure 3 (a) is applied on the unlabeled data, the performance usually has some substantial gains over the EMA-Teacher baselines, especially for the scenarios where the training is more difficult, *e.g.*, without self-pretraining or on 1% labels. This shows the importance of using mixup on the unlabeled data for an improved regularization. However, as discussed in Section 3.2, *pseudo mixup* could involve many noisy samples into training. On the other hand, *pseudo mixup+* of Figure 3 (b) can increase the performance of *pseudo mixup* constantly, by about 0.5 points, showing that removing those noisy samples does help. In addition, *probabilistic pseudo mixup* of Figure 3 (c) can further improve the performance of *pseudo mixup+* by 1-2 points in all cases. These results imply that those noisy samples do carry some useful information for SSL training, but their weights should be suppressed especially when their confidence scores are low. This data augmentation technique also effectively alleviates the training difficulty of semi-supervised vision transformers with weak inductive bias.

**Effect of Confidence Threshold** We ablate the effect of the confidence threshold $\tau$ of (2) in Table 4. We find that Semi-ViT is quite robust to the low confidence thresholds. One possible reason is that

Table 5: The ablation on the momentum decay of exponential moving average.

| Method (ViT-Base) | label | $m = 0$ | $m = 0.9$ | $m = 0.99$ | $m = 0.999$ | $m = 0.9999$ | $m = 0.99999$ |
|---|---|---|---|---|---|---|---|
| EMA-Teacher | 1% | ✗ | 23.5 | 49.3 | 63.1 | 65.3 | 59.7 |
| EMA-Teacher | 10% | 74.8 | 75.3 | 76.4 | 77.2 | 78.1 | 77.9 |
| Semi-ViT | 1% | 69.3 | 69.7 | 71.1 | 71.6 | 71.0 | 63.1 |
| Semi-ViT | 10% | 79.5 | 79.5 | 79.6 | 79.8 | 79.7 | 79.0 |

Table 6: The ablation on supervised fine-tuning.

| Method (ViT-Base) | label | epochs=0 | epochs=10 | epochs=50 | epochs=100 | epochs=200 |
|---|---|---|---|---|---|---|
| Supervised-ViT | 1% | - | 24.7 | 53.6 | 57.4 | 56.9 |
| Supervised-ViT | 10% | - | 66.3 | 72.9 | 73.7 | 73.2 |
| EMA-Teacher | 1% | 62.7 | 62.5 | 60.9 | 65.3 | 66.9 |
| EMA-Teacher | 10% | 76.5 | 74.2 | 77.7 | 78.1 | 78.2 |
| Semi-ViT | 1% | 69.7 | 69.8 | 70.4 | 71.0 | 70.9 |
| Semi-ViT | 10% | 79.3 | 79.4 | 79.6 | 79.7 | 79.6 |

Semi-ViT uses *probabilistic pseudo mixup*. When $\tau$ is low, the low-confidence samples will not hijack the training since their contributions depend on their confidence scores. These imply that the hyperparameter $\tau$ can possibly be removed ($\tau = 0$) in Semi-ViT. But the EMA-Teacher has a drop of 2-3 points when the confidence threshold is removed. The final choices of $\tau$ for different Semi-ViT models are shown in Table 13 in the appendix.

**Effect of Momentum Decay** Table 5 shows the effect of momentum decay $m$ in the EMA teacher. Note that when $m = 0$, the frameworks is reduced to the FixMatch. We can find Semi-ViT is robust to $m$. For 10% labels, Semi-ViT has very minor changes when $m$ decreases to 0, but the EMA-Teacher has a drop of 3.3 points. For 1% labels, the training is more challenging, and the choice of $m$ becomes more important. In this case, Semi-ViT has a drop of 2.3 points from $m = 0.999$ to $m = 0$, but the EMA-Teacher could fail when $m$ decreases to 0. The robustness of Semi-ViT to momentum decay $m$ is also attributed to the use of *probabilistic pseudo mixup*.

**Effect of Self-pretraining** The self-pretraining of MAE [25] has a substantial boost in performance, as seen in Table 3. For ViT-Base, MAE helps to improve by 6.2 and 9.2 points for the EMA-Teacher with and without *probabilistic pseudo mixup*, respectively. In addition, it helps to train the models in more challenging scenarios, *e.g.*, 1% labels. Without self-pretraining, the training fails to deliver good results on 1% labels. Notice that, even without pre-training, our Semi-ViT ("ProbPseudo Mixup" in Table 3) also achieves slightly better performance than the CNN counterparts: 70.9 of Semi-ViT-Small v.s. 67.1 of FixMatch-ResNet50 or 69.2 [55] of EMAN-ResNet50 [10] when trained from scratch for 100 epochs.

**Effect of Supervised Fine-tuning** is ablated in Table 6 by varying the number of *supervised fine-tuning* epochs. The rows of "Supervised-ViT" are the numbers of *supervised fine-tuning*, where the latter *semi-supervised fine-tuning* begins. Semi-ViT is still robust to the length of *supervised fine-tuning*, where the accuracy decrease is 0.4 (1.3) for 10% (1%) labels when *supervised fine-tuning* is removed. However, the performance of the EMA-Teacher decreases 1.7 (4.2) points. These show sufficient *supervised fine-tuning* does stabilize the training procedure, especially for less robust framework, *e.g.*, the EMA-Teacher. However, notice that *supervised fine-tuning* could sometimes hurt the performance if it is not sufficient (*e.g.*, epochs=10) for the EMA-Teacher.

**Other Self-pretraining Techniques** Beyond MAE, we also experiment on other self-pretraining techniques, including MoCo-v3 [16] and DINO [12], in Table 7. By comparing the fine-tuning results, DINO is close to MoCo-v3 for ViT-Base, but much better for ViT-Small, and both of them are better than MAE for ViT-Base, suggesting that DINO could be a better self-pretraining technique for smaller scales of ViT models. On top of the strong fine-tuning baselines, the *semi-supervised fine-tuning*, using the EMA-Teacher, still has nontrivial improvements for both DINO and MoCo-v3, *e.g.*, 5.8 (2.1) points on 1% (10%) labels for DINO-ViT-Base. In addition, the *probabilistic pseudo mixup* can further improve over the EMA-Teacher, independent of the self-pretraining algorithms. And the final Semi-ViT-Base of DINO is 2.1 (0.5) points better than that of MAE on 1% (10%) labels.

Table 7: Semi-ViT results with other self-pretraining techniques.

| Model | Pretrained | Method | 1% | 10% |
|-------|-----------|--------|-----|------|
| ViT-Small | MoCo-v3 [16] | finetune | 51.2 | 69.1 |
| | | EMA-Teacher | 61.9 | 72.3 |
| | | +ProbPseudo Mixup | 64.7 | 72.9 |
| ViT-Small | DINO [12] | finetune | 58.7 | 73.9 |
| | | EMA-Teacher | 66.3 | 76.3 |
| | | +ProbPseudo Mixup | 68.0 | 77.1 |
| ViT-Base | MoCo-v3 [16] | finetune | 66.3 | 74.5 |
| | | EMA-Teacher | 68.9 | 77.7 |
| | | +ProbPseudo Mixup | 72.3 | 79.2 |
| ViT-Base | DINO [12] | finetune | 65.0 | 76.0 |
| | | EMA-Teacher | 70.8 | 78.1 |
| | | +ProbPseudo Mixup | 73.1 | 80.2 |

Table 8: The results on ConvNeXt [45].

| Model | Upper-bound | Method | 10% |
|-------|------------|--------|------|
| ConvNeXt-T | 80.7 | supervised | 61.2 |
| | | EMA-Teacher | 70.4 |
| | | +ProbPseudo Mixup | 74.1 |
| ConvNeXt-S | 81.4 | supervised | 64.1 |
| | | EMA-Teacher | 71.7 |
| | | +ProbPseudo Mixup | 75.1 |

**Other Network Architectures** Although in this paper we mainly focus on ViT architectures, the proposed *probabilistic pseudo mixup* is not limited to them. We also try it for CNN architectures, *e.g.*, ResNet [27]. However, we find the direct use of the standard mixup does not improve fully-supervised ResNet performance, so will the *probabilistic pseudo mixup* for its SSL setting. Instead, we evaluate it on the recently proposed ConvNeXt [45], which uses mixup for improved results. Since the goal is not to fully reproduce the results of [45], all models are trained only for 100 epochs, including the supervised upper-bounds. The results in Table 8 demonstrate that *probabilistic pseudo mixup* is not limited to ViT, but also to CNN architectures, *e.g.*, with improvements of 3-4 points, suggesting it can be well generalized.

### 4.3 Comparison with the State-of-the-Art

Semi-ViTs are compared with the state-of-the-art semi-supervised learning algorithms in Table 9. When the model capacity is close, our Semi-ViT has shown much better results than the prior art, *e.g.*, MPL-RN-50 [53] v.s. Semi-ViT-Small, CowMix-RN152 [20] v.s. Semi-ViT-Base, S4L-RN50-4× [8] v.s. Semi-ViT-Large and SimCLRv2+KD-RN152-3×-SK [15] v.s. Semi-ViT-Huge. The only transformer based SSL method is SemiFormer [68], but it requires to use CNN as the teacher model and blend convolution and transformer modules together for good performance. However, our Semi-ViT is *pure* ViT based, without any additional parameters and architecture changes, and the Semi-ViT-Small model is already better than SemiFormer (77.1 v.s. 75.5). These comparisons support that Semi-ViT does advance the state-of-the-art of semi-supervised learning.

Scalability is an advantage of ViT, and we compare the scalability of Semi-ViT with previous works in Figure 1 (a) and (b). The comparison has shown that Semi-ViT can achieve better trade-off between model capacity and accuracy and can be scaled up more effectively than the prior art, SimCLRv2 [15]. For example, SimCLRv2 and PAWS [2] scale up the model usually in terms of network depth and width, and they seem to saturate when the model is of medium size, *e.g.*, around 300M parameters, but our Semi-ViT continues to improve steadily beyond that point.

Semi-ViT is also compared with the supervised state-of-the-art in Table 10. Our Semi-ViT-Huge is comparable with Inception-v4 [59], but with 100× annotation cost reduction, and comparable with ConvNeXt-L [45] (better than Swin-B [44]), but with 10× annotation cost reduction. These comparisons imply that Semi-ViT has great potential for labeling cost reduction.

Table 9: The comparison with the state-of-the-art SSL models.

| | Method | Architecture | Param | 1% | 10% |
|---|---|---|---|---|---|
| CNN | UDA [70] | ResNet-50 | 26M | - | 68.8 |
| | FixMatch [55] | ResNet-50 | 26M | - | 71.5 |
| | S4L [8] | ResNet-50 (4×) | 375M | - | 73.2 |
| | MPL [53] | ResNet-50 | 26M | - | 73.9 |
| | CowMix [20] | ResNet-152 | 60M | - | 73.9 |
| | EMAN [10] | ResNet-50 | 26M | 63.0 | 74.0 |
| | PAWS [2] | ResNet-50 | 26M | 66.5 | 75.5 |
| | SimCLRv2+KD [15] | RN152 (3×+SK) | 794M | 76.6 | 80.9 |
| Transformer | DINO [12] | ViT-Small | 22M | 64.5 | 72.2 |
| | SemiFormer [68] | ViT-S+Conv | 42M | - | 75.5 |
| | Semi-ViT (ours) | ViT-Small | 22M | 68.0 | 77.1 |
| | Semi-ViT (ours) | ViT-Base | 86M | 71.0 | 79.7 |
| | Semi-ViT (ours) | ViT-Large | 307M | 77.3 | 83.3 |
| | Semi-ViT (ours) | ViT-Huge | 632M | 80.0 | 84.3 |

Table 10: The comparison with the state-of-the-art fully supervised models.

| | Model | Param | Data | top-1 | top-5 |
|---|---|---|---|---|---|
| CNN | ResNet-50 [27] | 26M | ImageNet | 76.0 | 93.0 |
| | ResNet-152 [27] | 60M | ImageNet | 77.8 | 93.8 |
| | DenseNet-264 [32] | 34M | ImageNet | 77.9 | 93.9 |
| | Inception-v3 [60] | 24M | ImageNet | 78.8 | 94.4 |
| | Inception-v4 [59] | 48M | ImageNet | 80.0 | 95.0 |
| | ResNeXt-101 [72] | 84M | ImageNet | 80.9 | 95.6 |
| | SENet-154 [31] | 146M | ImageNet | 81.3 | 95.5 |
| | ConvNeXt-L [45] | 198M | ImageNet | 84.3 | - |
| | EfficientNet-L2 [61] | 480M | ImageNet | 85.5 | 97.5 |
| Transformer | ViT-Huge [18] | 632M | JFT+ImageNet | 88.6 | - |
| | DeiT-B [63] | 86M | ImageNet | 81.8 | - |
| | Swin-B [44] | 88M | ImageNet | 83.3 | - |
| | MAE-ViT-Huge [25] | 632M | ImageNet | 86.9 | - |
| | Semi-ViT-Huge (ours) | 632M | 1%ImageNet | 80.0 | 93.1 |
| | Semi-ViT-Huge (ours) | 632M | 10%ImageNet | 84.3 | 96.6 |

### 4.4 Other Datasets

The generalization of Semi-ViT is evaluated on datasets including Food-101 [9], iNaturalist [30] and GoogleLandmark [52]. Since these datasets are beyond ImageNet, we assume that the ImageNet dataset is available and the model is already supervised pretrained on ImageNet, and then the model is finetuned to different target datasets with a few labels. The results are shown in Table 11. On these datasets, our Semi-ViT can improve over the fine-tuning baselines by 13-21 (7-10) points on 1% (10%) labels. Note that on Food-101, Semi-ViT on 1% (10%) labels is close to fine-tuning baseline on 10% (100%) labels, *i.e.*, 82.1 v.s. 84.5 (91.3 v.s. 93.1), indicating that using Semi-ViT can help to save annotation costs by about 10 times on this dataset.

## 5 Related Work

Semi-supervised learning has a long history of research [78, 13]. The recent works can be roughly clustered into two groups, consistency-based [39, 62, 50, 70, 66] and pseudo-labeling based [40, 55, 53, 10]. Consistency-based methods usually add some noise to the input or the model, and then enforce their feature or probability outputs to be consistent. For example, to construct two outputs for later consistency regularization, Π-model [39] adds noise to the model weights using dropout [57], Mean-teacher [62] builds a teacher model by EMA updated from the student model, and UDA [70] applies a weak and a strong data augmentation to the input. On the other hand, the idea of pseudo-labeling or self-training can be traced back to [34, 49], which uses model predictions as hard pseudo labels to guide the learning on unlabeled data. This idea becomes popular in SSL recently [40, 55, 53, 10, 71], and some theoretical explanations are available [76, 24]. In the offline pseudo labeling [40, 71], the model used to generate pseudo labels is usually frozen or updated once

Table 11: The Semi-ViT-Base results on other datasets.

| Dataset | # train/test | # class | Method | 1% | 10% | 100% |
|---------|-------------|---------|--------|-----|------|------|
| Food-101 [9] | 75.7K/25.2K | 101 | Finetune | 60.9 | 84.5 | 93.1 |
| | | | Semi-ViT | 82.1 | 91.3 | - |
| iNaturalist [30] | 265K/3K | 1010 | Finetune | 19.6 | 57.3 | 81.2 |
| | | | Semi-ViT | 32.3 | 67.7 | - |
| GoogleLandmark [52] | 200K/15.6K | 256 | Finetune | 45.3 | 74.0 | 91.5 |
| | | | Semi-ViT | 61.0 | 81.0 | - |

in a while during training, *e.g.*, at the end of every training epoch, but for online pseudo-labeling [55, 10] the teacher model is updated continuously along with the student. Beyond classification, pseudo-labeling has also achieved promising progress in more challenging tasks, *e.g.*, object detection [56, 43, 67]. Our Semi-ViT also falls into the category of online pseudo-labeling.

Mixup [75] is an effective data augmentation technique, which interpolates the input samples and their labels linearly and performs vicinal risk minimization. It has been successfully used in image classification and some other domains, *e.g.*, generative adversarial networks [47], sentence classification [23], etc. Other variants have also been developed, *e.g.*, Manifold Mixup [65] that mixes up in the feature space or CutMix [74] which cuts a patch from one image and pastes it into another one. Mixup has also been successfully adopted in self-supervised learning [37, 41] and semi-supervised learning [7, 66, 6]. Although [7, 66, 6] also use mixup for SSL, they have differences with our *probabilistic pseudo mixup*: 1) they are consistency-based SSL framework, but ours is pseudo-labeling based; 2) their mixup ratio is random sampled, but ours depends on the pseudo label confidence; 3) they have only shown successes on small CNN architectures and small datasets, *e.g.*, CIFAR [38] and SVHN [51], but our successes are built on various scales of transformer architectures and large-scale datasets, *e.g.*, ImageNet [54], INaturalist [30], GoogleLandmark [52], etc.

## 6 Conclusion

In this paper, we propose Semi-ViT for vision transformers based semi-supervised learning. This is the first time that *pure* vision transformers can achieve promising results on semi-supervised learning and even surpass the previous best CNN based counterparts by a large margin. In addition, Semi-ViT inherits the scalable benefits from ViT, and the larger model leads to smaller gap to the fully supervised upper-bounds. This has shown to be a promising direction for semi-supervised learning. And the advantages of Semi-ViT can be well generalized to other datasets, suggesting potentially broader impacts. We hope these promising results could encourage more efforts in semi-supervised vision transformers.

**Limitations**  Our paper only considers the standard semi-supervised classification settings where the full dataset, *e.g.*, ImageNet, is downsampled to smaller scales, and not the advanced settings where full ImageNet is used as labeled and additional data, *e.g.*, ImageNet-21K, is used as unlabeled. And we have only evaluated our approach on the classification task. It is unclear whether the same conclusion holds in the case of a more advanced classification setting and more challenging tasks, *e.g.*, detection or segmentation.

**Potential Negative Social Impacts**  Semi-ViT has shown that strong models can be obtained with only a few labels, *e.g.*, 1%. This increases the good AI models accessibility to anyone, which could potentially lead to their inappropriate use.

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
