# A Implementation Details

**Dataset Sampling**    We sample 1% (10%) images per class from the datasets we use for the semi-supervised learning experiments of 1% (10%) labels. For example, on ImageNet, the number of sampled images is 12,820 (128,118) in total, for 1% (10%) labels.

**ViT Architectures**    We use exactly same architecture as the standard ViT [18]. For the position tokens, we use the learnable positional embedding when the model is self-pretrained, but the sine-cosine version of positional embedding (non-learnable) when not self-pretrained since we find it leads to better results than the learnable one. For all architectures, the classifier is built on top of the average pooling of the encoder output, except for ViT-Huge where we find the classifier on top of the output of the class token leads to higher performances.

**Data Augmentation**    We use the common data augmentations of `RandomResizedCrop`, `RandomHorizontalFlip`, `RandAugment('m9-mstd0.5-inc1')` [17] and `RandomErasing` [77] on the labeled data. The same augmentations are also used on the unlabeled data as the strong augmentation in EMA-Teacher, whereas the weak augmentations are `RandomResizedCrop`, `RandomHorizontalFlip` and `ColorJitter(0.4)`. We do not extensively explore the space of data augmentation for weak and strong augmentations. The center 224×224 crop is used at inference.

**Self-supervised Pretraining**    We directly use the pretrained models from MAE [25], DINO [16] and MoCo-v3 [12]. The final Semi-ViT-Small is with DINO self-pretraining.

**Supervised Fine-tuning Settings**    The settings for the stage of *supervised fine-tuning*, with and without self-pretraining, are shown in Table 12. These settings are mainly following those of finetuning and learning from scratch in [25], with some minor modifications. We also use the linear learning rate scaling rule [21]: $lr = base\_lr \times batchsize/256$. When supervised fine-tuning on 1% data, we find the performances are bad when the regularization is strong, hence we do not use mixup/cutmix, drop path and random erasing, in that case, except ViT-Small.

**Semi-supervised Fine-tuning Settings**    The settings for the stage of *semi-supervised fine-tuning*, with and without self-pretraining, are shown in Table 13. When semi-supervised fine-tuning small models (ViT-Small/Base) on 1% data, we find the performances are bad when the regularization is strong, hence we do not use mixup/cutmix on labeled data, and drop path and random erasing on both labeled and unlabeled data, in that case. The linear learning rate scaling rule is: $lr = base\_lr \times batchsize_l/256$, where $batchsize_l$ is the batch size of the labeled data.

**Confidence Threshold**    The confidence thresholds of Semi-ViT are shown in Table 13, even though sometimes those are not optimal according to the ablation study in Table 4.

**Computing Resources**    We run all experiments on V100 GPUs of 32G memory. For Semi-ViT-Small/Base/Large/Huge, we use 8/8/16/32 GPUs for training 100/100/100/50 epochs, which takes about 19/31/61/115 hours.

**Random Seeds and Error Bar**    Since some of the experiments are expensive to run, e.g., Semi-ViT-Huge. We only test the randomness on Semi-ViT-Base models. We sample three different 10%/1% subsets of ImageNet and repeat the experiments for three times with different random seeds. When self-pretrained by MAE, the accuracy is $79.71 \pm 0.037$ ($70.95 \pm 0.029$) for 10% (1%) labels. When not self-pretrained, the accuracy is $73.44 \pm 0.065$ for 10% labels. The results have shown that Semi-ViT is quite robust to different random seeds and different subsets of the samples. To keep consistency, all experiments in the paper are run with the same ImageNet subset and the same random seed.

**Settings on Other Datasets**    The model is ViT-Base on the datasets of Food-101 [9], iNaturalist [30] and GoogleLandmark [52]. The *supervised fine-tuning* settings are almost the same as those with self-pretraining in Table 12, with the differences: 1) we search the optimal base learning rate from $\{1e^{-4}, 2.5e^{-4}, 5e^{-4}, 1e^{-3}\}$ and the optimal layer-wise learning rate decay from $\{0.65, 0.75\}$;

Table 12: Supervised fine-tuning settings with and without self-pretraining.

| config | 10% labels | 1% labels | 10% labels ( scratch) |
|---|---|---|---|
| optimizer | AdamW | AdamW | AdamW |
| base learning rate | 1e-4 (S), 2.5e-4 (B) 1e-3 (L/H) | 1e-4 (S), 5e-5 (B) 1e-3 (L), 0.01 (H) | 1e-4 |
| weight decay | 0.05 | 0.05 | 0.3 |
| optimizer momentum | $\beta_1, \beta_2$=0.9, 0.999 | $\beta_1, \beta_2$=0.9, 0.999 | $\beta_1, \beta_2$=0.9, 0.95 |
| layer-wise lr decay [5] | 0.65 (S/B), 0.75 (L/H) | 0.65 (S/B), 0.75 (L/H) | 1.0 |
| batch size | 512 (S/B/L), 256 (H) | 512 (S/B/L), 128 (H) | 1024 |
| learning rate schedule | cosine decay | cosine decay | cosine decay |
| warmup epochs | 5 | 5 | 50 |
| training epochs | 100 (S/B), 50 (L/H) | 100 (S/B), 50 (L/H) | 500 |
| label smoothing [60] | 0.1 | 0.1 | 0.1 |
| mixup [75] | 0.8 | 0.8 (S), 0 (B/L/H) | 0.8 |
| cutmix [74] | 1.0 | 1.0 (S), 0 (B/L/H) | 1.0 |
| drop path [33] | 0.1 (S/B/L) 0.2 (H) | 0.1 (S), 0 (B/L/H) | 0.1 |
| random erasing [77] | 0.25 | 0.25 (S), 0 (B/L/H) | 0.25 |

Table 13: Semi-supervised fine-tuning settings with and without self-pretraining.

| config | 10% labels | 1% labels | 10% labels ( scratch) |
|---|---|---|---|
| optimizer | AdamW | AdamW | AdamW |
| base learning rate | 2e-4 (S), 1e-3 (B) 2e-3 (L), 2.5e-3 (H) | 5e-4 (S), 1e-3 (B/L) 5e-3 (H) | 1e-3 |
| weight decay | 0.05 | 0.05 | 0.05 |
| optimizer momentum | $\beta_1, \beta_2$=0.9, 0.999 | $\beta_1, \beta_2$=0.9, 0.999 | $\beta_1, \beta_2$=0.9, 0.999 |
| layer-wise lr decay [5] | 0.75 | 0.75 | 0.85 |
| batch size (labeled) | 128 | 128 (S/B/L), 64 (H) | 128 |
| learning rate schedule | cosine decay | cosine decay | cosine decay |
| confidence threshold | 0.5 (S/B), 0.6 (L/H) | 0.6 | 0.5 |
| warmup epochs | 5 | 5 | 5 |
| training epochs | 100 (S/B/L), 50 (H) | 100 (S/B/L), 50 (H) | 100 |
| label smoothing [60] | 0.1 | 0.1 | 0.1 |
| mixup [75] | 0.8 | 0.8 | 0.8 |
| cutmix [74] | 1.0 | 1.0 | 1.0 |
| drop path [33] | 0.1 (S/B/H), 0.2 (L) | 0 (S/B), 0.1 (L), 0.05 (H) | 0.1 |
| random erasing [77] | 0.25 | 0 (S/B), 0.25 (L/H) | 0.25 |

2) we enable mixup/cutmix, drop path and random erasing for 1% experiments (except iNaturalist), with the same values of those for 10% experiments. The *semi-supervised fine-tuning* settings are almost the same as those with self-pretraining in Table 13, with the difference that we set the EMA momentum decay $m = 0.999$ instead of $m = 0.9999$ as in the ImageNet experiments, since these datasets are smaller and need faster EMA update rate.