# OpenReview forum: "Semi-supervised Vision Transformers at Scale"
_NeurIPS.cc/2022/Conference — NeurIPS 2022 Accept_

### Official Review · Reviewer_Uvx3 · 2022-07-09

**Rating:** 6
**Confidence:** 4
**Soundness:** 4 excellent
**Presentation:** 3 good
**Contribution:** 3 good

**Summary:**

This paper proposes a semi-supervised framework for vision transformers. In which, author introduces two techniques to improve the robustness and performance of ViT in semi-supervised learning. They are 1. EMA-teach network update which is the moving average of the student network. 2. Probabilistic Pseudo Mixup which is a novel mix-up method under pseudo-labelling based SSL framework.

**Questions:**

1.	Under a more general semi-supervised learning set-up, how does the proposed method work when labelled data and unlabelled data are from different datasets?

**Limitations:**

1.	The large-scaled self-supervised pre-training (MAE) may generate more carbon emission.

**Strengths And Weaknesses:**

Strengths:

1.	This paper is well written, and the core idea is easy to understand. The proposed method and formulate is clean, straightforward, and easy to re-implement.

2.	The proposed method effectively improves the semi-supervised training for ViT. Compared to the baseline, both EMA-teacher updating, and Probabilistic Pseudo Mixup achieve significant improvement.

3.	The Probabilistic Pseudo Mixup is novel, which provide a new direction to employ mix-up in ViT under pseudo-labelling based SSL framework.

4.	The experimental result is remarkable. Compared to fully supervised finetuning after MAE, this paper is only 2% lower with only 10% imageNet data. In addition, the proposed method works well under various self-supervised pretraining pipelines.

Weaknesses:

1.	It will be good to show more ablation study over some hyper-parameters, such as the momentum decay and confidence score.

---

> ### Author Response · Authors · 2022-08-01
> **Response to Reviewer Uvx3**
>
> Thanks for the constructive review! We provide the detailed responses to each question as below.
>
> **Q1: It will be good to show more ablation study over some hyper-parameters, such as the momentum decay and confidence score.**
>
> A: Thanks for the suggestion. We are adding more ablation studies.
>
> |          Method         | data | $\tau$=0.3 | $\tau$=0.4 | $\tau$=0.5 | $\tau$=0.6 |
> |:---------------------:|:----------------:|:----------------:|:----------------:|:----------------:|:----------------:|
> |       Semi-ViT-Base       |      10%      |       79.5      |       79.7     |      79.7     |       79.6     |
> |       Semi-ViT-Base       |      1%        |       71.4      |       71.3     |      71.3     |       71.0     |
>
> First, we show the ablation on confidence threshold $\tau$ in the above table. It can be found that our Semi-ViT is quite robust to the filtering threshold. One possible reason is that we use probabilistic pseudo-mixup, even when some samples are filtered out by the threshold, they still can contribute to the final loss, and their contributions depend on their confidence scores, so the low-confidence samples won’t hijack the training. In the submission, we used $\tau$=0.5 ($\tau$=0.6) for Semi-ViT-Base on 10% (1%) labels.
>
> |          Method         | data | $m$=0.99 | $m$=0.999 | $m$=0.9999 | $m$=0.99999 |
> |:---------------------:|:----------------:|:----------------:|:----------------:|:----------------:|:----------------:|
> |       Semi-ViT-Base       |      10%      |       79.6      |       79.8     |      79.7     |       79.0     |
>
> Next, we show the ablation on the momentum decay rate $m$ in the above table. We can find that our Semi-ViT is also quite robust to the momentum decay rate. Somewhere between 0.99 to 0.9999 works fine.
>
> These two experiments support that our Semi-ViT are robust to hyperparameters.
>
> **Q2: Under a more general semi-supervised learning set-up, how does the proposed method work when labelled data and unlabelled data are from different datasets?**
>
> A: It is an interesting idea to have unlabelled data from a different dataset. However, this is a more challenging task. It requires to have a more robust filtering mechanism, since using another dataset will introduce the problems of domain gap and category differences. Thus, probably new algorithms need to be developed. In fact, this is a subtask of SSL, and some papers specifically work on this. Due to the limited time, we are unable to come up new algorithm for this, but it will be remained as an interesting future work.
>
> **Q3: The large-scaled self-supervised pre-training (MAE) may generate more carbon emission.**
>
> A: Yes, the self-supervised pretraining is usually expensive. However, we simply just reused the pretrained models provided by those works. So in our experiments, we didn’t generate those carbon emission caused by self-pretraining.

---

### Official Review · Reviewer_zFZY · 2022-07-11

**Rating:** 4
**Confidence:** 4
**Soundness:** 3 good
**Presentation:** 3 good
**Contribution:** 2 fair

**Summary:**

This work proposes a three steps training framework for pure ViTs, including un/self-supervised pre-training, followed by supervised fine-tuning, and finally semi-supervised fine-tuning. EMA and a probabilistic pseudo mixup mechanism are used and results are competitive.

**Questions:**

Except that ViTs are sometimes data hungry, what are the main differences between ViTs and CNNs that need the community to pay spatial attention to achieve good SSL results? Does the proposed method also benefit CNNs models?

**Limitations:**

The necessity to design spatial methods for SSL on ViTs  needs to be clarified.

**Strengths And Weaknesses:**

+ The paper is well written and easy to follow.
+ First to use pure ViT for SSL.

- The proposed training pipeline are not new compared with former works, such as [14].
- The improvements are based on existing works (i.e. EMA-Teacher) that are easy to come up with in the semi-supervised domain.

---

> ### Author Response · Authors · 2022-08-01
> **Response to Reviewer zFZY**
>
> Thanks for the valuable review! We provide the detailed responses to each question as below.
>
> **Q1: The proposed training pipeline are not new compared with former works, such as [14].**
>
> A: The current popular pipelines for SSL are somehow similar, as we described in Section 2.1. However, our pipeline does have some differences from [14], which uses knowledge distillation in their final stage, instead of a semi-supervised fine-tuning as we use. We are not claiming that the three-stage pipeline is technically novel, and the goal of our paper is not to argue which pipeline is better. We just found our three-stage pipeline enables to have stable training of Semi-ViT and reduce the hyperparameter tuning. It is an important recipe to have stable training and good results for semi-supervised ViT. The extra ablation studies we provide to other reviewers do support that.
>
> **Q2: The improvements are based on existing works (i.e. EMA-Teacher) that are easy to come up with in the semi-supervised domain.**
>
> A: Perhaps in hindsight it may seem like an easy change. The choice does not seem as easy when faced with hundreds of options for techniques to apply to SSL and the prospect of having to try them exhaustively. This stands in contrast with other choices, for instance SemiFormer’s use of FixMatch [60] which yields disappointing performance. We view the simplicity of the resulting method, and its use of established techniques, as a strength rather than a limitation.
>
> What we do claim novelty for is the probabilistic pseudo mixup, which has some appealing properties. For example, although some samples don’t pass the confidence threshold, they still can contribute to the final loss, and their contributions depend on their confidence scores, so the low-confidence samples won’t hijack the training. As a result, our Semi-ViT is robust to confidence threshold, as also pointed out in the response to Q2 of Reviewer-mtWa.
>
> **Q3: What are the main differences between ViTs and CNNs that need the community to pay spatial attention to achieve good SSL results?**
>
> A: Our paper shows that three aspects are key to achieving results such as 80% top-1 accuracy with 1% of ImageNet labels: training pipeline, SSL framework, and data augmentation. And we have provided good recipes for them, which could ease the overhead of the future efforts in this direction.
>
> **Q4: Does the proposed method also benefit CNNs models?**
>
> A: Yes, as shown in our Table 5 (e.g. ConvNeXt). Those experiments illustrate the generalization properties of the proposed techniques in our paper.

---

### Official Review · Reviewer_mtWa · 2022-07-11

**Rating:** 8
**Confidence:** 5
**Soundness:** 4 excellent
**Presentation:** 4 excellent
**Contribution:** 3 good

**Summary:**

Recently pseudo-labeling demonstrated powerful results in many domain, including object detection, speech and image recognition, NLP and others. Current paper continues series of works on pseudo-labeling in context of ViT architecture and understanding different aspects of successful pipeline for ViT models with respect to scaling and reducing supervised data. First, authors proposed probabilistic mixup which allows to use filtered pseudo-labeled data to augment non-filtered pseudo-labeled data: weights of mixup are not sampled from beta distribution but pseudo-label score defines them. This scheme is shown with many experiments and ablations to be very effective and give consistent significant improvement. Second, authors confirm that FixMatch is unstable scheme of training in both cases having or not the self-supervised pretraining in the regime of low supervision (1% or 10% of ImageNet is used as labeled data). Third, authors demonstrated that self-supervised pretraining is complementary to pseudo-labeling and combination together improves results by a lot especially in 1% labeled data setting (this result was shown in several domains too, e.g. speech recognition). Finally authors show great scalability of pseudo-labeling for ViT models (with self-supervised pretraining, supervised finetuning and then EMA pseudo-labeling finetuning) and reach impressive results with only 1% labeled data of ImageNet compared to ImageNet supervised baselines.

**Questions:**

The paper is very well written with strong results, clear explanation of experiments and settings. I have very little suggestion on improving the paper:
- Could author do sorting of references, so that their numbers appear in the order they first mentioned in the paper? In this case much simpler to find works from the introduction section.
- References are great, thanks for this! Only would be great to have extra references to other domains, like speech recognition and NLP, where pseudo-labeling is actively developed too, including EMA and stabilization of training for online pseudo-labeling (model is trained with continuously updated teacher). Let me know if you need particular pointers as I am familiar with these works. There are also 2 recent theoretical papers on explaining why pseudo-labeling works (both teacher-student and online versions) [1,2].
- Extra observations on divergence and instability of FixMatch are very important to have for future pseudo-labeling development. I like details provided in Sec 2.2 which gives proper combined references to prior works and observations. I would like to see here also references to several speech recognition works and have even stronger overview on the instability training. Yes, it is a bit different variant of FixMatch people use in speech, but the same observation on usage this type of teacher holds - people observe divergence and huge instability especially (if teacher and student share weights) with very little supervision (see [3] paper Fig.1) and they propose either history cache [3] or EMA [4,5] to stabilize the training.
- line 119 - use set difference not subtraction
- Maybe I missed but I didn't notice reference to ImageNet
- Authors did comprehensive experiments showing what is the contribution of self-supervised pretraining, probabilistic mixup in different scenarios, different models, different pretraining. Only one thing remains questions for me: how does the filtering really affect the training? Could it be that with probabilistic mixup we even can avoid it and reduce number of hyperparameters (e.g. have ablation in Table 3 where filtering is not used)? (Just in context, in speech recognition absence of filtering works very well, and filtering techniques so far show only marginal gain for CTC models, where there is no problem for filtering long/short sequences as in seq2seq type of loss). I also didn't find the final filtering threshold in the Appendix, only the set which was grid searched.
- [Extra question]. I wonder how number of epochs of supervised finetuning influences the overall EMA pseudo-labeling convergence? Are there different scenarios for 1% and 10% labeled data? (See e.g. such study in [3, 4, 5] for speech).

Overall, thanks authors for the deep investigation!

[1] Zhang, S., Wang, M., Liu, S., Chen, P.Y. and Xiong, J., 2022. How does unlabeled data improve generalization in self-training? A one-hidden-layer theoretical analysis. ICLR 2022.

[2] He, H., Yan, H. and Tan, V.Y., 2021. Information-theoretic generalization bounds for iterative semi-supervised learning. arXiv preprint arXiv:2110.00926.

[3] Likhomanenko, T., Xu, Q., Kahn, J., Synnaeve, G. and Collobert, R., 2020. slimipl: Language-model-free iterative pseudo-labeling. Interspeech 2021.

[4] Manohar, V., Likhomanenko, T., Xu, Q., Hsu, W.N., Collobert, R., Saraf, Y., Zweig, G. and Mohamed, A., 2021, December. Kaizen: Continuously improving teacher using exponential moving average for semi-supervised speech recognition. In 2021 IEEE Automatic Speech Recognition and Understanding Workshop (ASRU) (pp. 518-525). IEEE.

[5] Higuchi, Y., Moritz, N., Roux, J.L. and Hori, T., 2021. Momentum pseudo-labeling for semi-supervised speech recognition. Interspeech 2021.


**Limitations:**

Limitations are listed in the conclusion section.

**Strengths And Weaknesses:**

**Strengths**
- Very well, clearly written paper with all necessary details and deep explanations
- Comprehensive experimental study of pseudo-labeling for ViT and proper ablations showing consistent results across the board
- New idea of probabilistic mixup which gives consistent experimental improvement across the board for different scenarios and pipelines
- Ablations on FixMatch confirming training instability for low supervision setting
- Ablations showing complementary property of pseudo-labeling and self-supervised pretraining
- Impressive results with 1% labeled data only
- Demonstration of scaling property for pseudo-labeling for ViT architecture

**Weaknesses**
- [not important] Absence of some recent literature on theoretical justification of pseudo-labeling and similar EMA study and stability in other domains (see Questions section on more details)
- Absence of investigation at what extent filtering is important in the pipeline (regarding that mixup with filtered data helps) - this could be another baseline for probabilistic mixup justification
- [maybe future work?] Absence of study how many epochs of supervised training / supervised finetuning is needed before starting EMA pseudo-labeling process.

---

> ### Author Response · Authors · 2022-08-01
> **Response to Reviewer mtWa**
>
> Thanks for this constructive review and the recognition of our work! We provide the detailed responses to each question as below.
>
> **Q1: Absence of some recent literature on theoretical justification of pseudo-labeling and similar EMA study and stability in other domains**
>
> A: We thank the reviewer for providing pointers to those papers, which are on a domain outside of our expertise. We will address them correctly in our camera-ready.
>
> **Q2: Absence of investigation at what extent filtering is important in the pipeline (regarding that mixup with filtered data helps) - this could be another baseline for probabilistic mixup justification**
>
> A: Does the reviewer mean what’s the effect of filtering in our pipeline? The filtering depends on the confidence threshold. And we are adding more ablation studies on the filtering threshold $\tau$ as below. It can be found that our Semi-ViT is quite robust to the filtering threshold. One possible reason is that we use probabilistic pseudo-mixup, even when some samples are filtered out by the threshold, they still can contribute to the final loss, and their contributions depend on their confidence scores, so the low-confidence samples won’t hijack the training. In fact, when $\tau$=0.3, 99% unlabeled samples pass the threshold, and we can almost say there is no filtering in this case. This observation is somehow aligned with the observation in the speech domain as mentioned by the reviewer. This is kind of interesting and maybe it is possible we can remove filtering in our Semi-ViT as well. But more experiments are needed to make that conclusion. In the submission, we used $\tau$=0.5 ($\tau$=0.6) for Semi-ViT-Base on 10% (1%) labels.
>
> |          Method         | data | $\tau$=0.3 | $\tau$=0.4 | $\tau$=0.5 | $\tau$=0.6 |
> |:---------------------:|:----------------:|:----------------:|:----------------:|:----------------:|:----------------:|
> |       Semi-ViT-Base       |      10%      |       79.5      |       79.7     |      79.7     |       79.6     |
> |       Semi-ViT-Base       |      1%        |       71.4      |       71.3     |      71.3     |       71.0     |
>
> **Q3: Absence of study how many epochs of supervised training / supervised finetuning is needed before starting EMA pseudo-labeling process.**
>
> A: Thanks for the suggestion, and we are adding these experiments. The below table is the accuracies of supervised finetuning. The default setting of Semi-ViT is using 100 supervised training epochs. We can find when training less epochs, the finetuning results will be slightly worse for 10% labels, but much worse for 1% labels. When training longer, the accuracies usually have minor drops.
>
> |          Method         | data | supervised epoch=50 | supervised epoch=100 | supervised epoch=200 |
> |:---------------------:|:----------------:|:----------------:|:----------------:|:----------------:|
> |       ViT-Base       |      10%      |       72.9      |       73.7     |      73.2     |
> |       ViT-Base       |      1%        |       53.6      |       57.4     |      56.9    |
>
> Next, we show the accuracies of Semi-ViT starting from different numbers of epochs of supervised finetuning. We can find the Semi-ViT is robust enough (with minor differences), although the supervised finetuning accuracies have some substantial differences, e.g. for 1% labels. These experiments show the robustness of our three-stage pipeline.
>
> |          Method         | data | supervised epoch=50 | supervised epoch=100 | supervised epoch=200 |
> |:---------------------:|:----------------:|:----------------:|:----------------:|:----------------:|
> |       Semi-ViT-Base       |      10%      |       79.6      |       79.7     |      79.6     |
> |       Semi-ViT-Base       |      1%        |       70.4      |       71.0     |      70.9    |
>
> **Q4: Could author do sorting of references, so that their numbers appear in the order they first mentioned in the paper?**
>
> A: Our references are sorted by the last names of the authors, which is the most common in computer vision community. We will check what is the requirement of NeurIPS and will comply with it.
>
> **Q5: Missing references**
>
> A: Please see Q1
>
> **Q6: Extra observations on divergence and instability of FixMatch**
>
> A: It is interesting to know the same observations have also been seen in other domains, e.g. speech. These extra observations do provide additional support that FixMatch is unstable in some scenarios, and EMA-Teacher could be a better choice in general. We will add some discussion and references on speech too, as suggested in this comment.
>
> **Q7: line 119 - use set difference not subtraction**
>
> A: Thanks for pointing this out. Will fix it.
>
> **Q8: reference to ImageNet**
>
> A: ImageNet reference is [46]
>
> **Q9: how does the filtering really affect the training?**
>
> A: See Q2.
>
> **Q10: I wonder how number of epochs of supervised finetuning influences the overall EMA pseudo-labeling convergence?**
>
> A: See Q3.

---

> > ### Comment · Reviewer_mtWa · 2022-08-07
> > **Thanks for additional ablations and clarifications**
> >
> > Dear authors,
> >
> > Thanks for additional ablations and discussions. Please find below some thoughts and suggestions I would like to see in the final revision:
> > - (regarding ablation on $\tau$). Thanks for these quick experiments! Very well that algorithm is robust. Still the most fair experiment (**also to justify more strongly proposed probabilistic mixup**) will be having $\tau=0$ both for the baseline and Semi-ViT. I suspect that probabilistic mixup is doing proper regularization here and will perform much better than the baselines. Thus it can be viewed as alternative to uncertainty estimation / confidence filtering / weighting methods which people use in SSL to be robust to noise in pseudo-labels. That is why I wanna you to confirm that probabilistic mixup can be viewed even wider as done right now in the paper and we can remove one hyperparameter (simplification, which is good). However, it is not critical for my decision on the paper at current stage of discussion.
> > - (regarding your ablations on the EMA decay factor) Add details when EMA accumulation starts (is it before first batch of unlabeled data is used or right away from this first batch?). Your experiments show that model should change not too slow and probably not too fast. I wonder to what limit you can push EMA decay, say 0.9, 0.1? I believe in 1% sup. data scenario it could be less stable and larger decay factor is needed compared to 10%.
> > - (regarding how many supervised epoch we do before pseudo-labels are involved) Thanks for quick ablations. With respect to 1% setting: seems that we tends to overfit to the labeled set. Do you know what is the best supervised baseline we can have here for both 1% and 10% settings (to have understanding when we start pseudo-label training)? I suggest to add additional supervised model quality for these 50, 100, 200 epochs. This could give a hint for future research how the quality of model when we start pseudo-labeling correlates with the final performance. Could you extend a bit this table in final revision to have 1, 10 epochs too? I wonder to what limit we can push this to be able to bootstrap model with pseudo-labels.

---

> > > ### Author Response · Authors · 2022-08-07
> > > **Follow-up responses**
> > >
> > > Dear Reviewer,
> > >
> > > Thanks for these valuable suggestions! The responses are as below.
> > >
> > > * For confidence threshold $\tau$, we totally agree this comment. We are running the experiments for other thresholds, e.g. $\tau=0$, and also on the baselines without using probabilistic pseudo mixup. We believe, without probabilistic pseudo mixup, the model will be much less robust to $\tau$.
> > > * We will also run the experiments beyond 0.99, e.g. 0.9 or even 0.0, and also on 1% labels. We agree that the experiments on 1% will be less robust on $m$, since the training on 1% labels is less stable.
> > > * We think the best time to start semi-supervised finetuning is when the performances saturate at the stage of supervised finetuning, e.g. 100 (50) epochs for small/base (large/huge) models. But it is possible that semi-supervised finetuning could start earlier. Due to the limited time, we could only run the experiments for 50, 100, 200 epochs. We will run the experiments for 1 or 10 epochs of supervised finetuning.
> > >
> > > Since it is expensive to extensively run these ablation studies, we can't finish all these experiments during the rebuttal period. We will try out best to finish them as soon as possible. And these ablation experiments and discussion will be presented in the final version.
> > >
> > > Please let us know if you have further questions.

---

### Official Review · Reviewer_Nfe1 · 2022-07-18

**Rating:** 4
**Confidence:** 3
**Soundness:** 2 fair
**Presentation:** 2 fair
**Contribution:** 2 fair

**Summary:**

This paper proposed Semi-ViT, a semi-supervised learning approach for vision transformers. The proposed method consists of three stages: first un/self-supervised pre-training, followed by supervised fine-tuning, and finally semi-supervised fine-tuning. At the semi-supervised fine-tuning.

At the semi-supervised fine-tuning stage, Semi-ViT adopts two techniques: an exponential moving average (EMA)-Teacher framework and a probabilistic pseudo mixup mechanism, to improve the performance.

Semi-ViT, achieves comparable or better performance than the CNN counterparts in the semi-supervised classification setting.

The authors also show promising scaling up experiments, such as: Semi-ViT-Huge achieves an impressive 80% top-1 accuracy on ImageNet using only 1% labels, which is comparable with Inception-v4 using 100% ImageNet labels.


**Questions:**

As in "Strengths And Weaknesses"

**Limitations:**

yes.

**Strengths And Weaknesses:**

Strength:
The proposed method is clearly written. It can be understood easily.

Weakness:
Lack of novelty. The proposed three-stage training, EMA teacher, and the probabilistic Pseudo Mixup are all well-known techniques. (the specific techniques in probabilistic Pseudo Mixup is new, but Pesudo Mixup is very natural). All the experimental results are as expected. I did not learn much new here.
For comparisons in Figure 1 (a,b), I'm not sure whether the baseline methods (SimCLRv2, PAWS, EMAN) are also trained in a three-stage manner (e.g., the third semi-supervised stage for SimCLRv2 can be just standard semi-supervised learning with EMA teacher). If not, the merge of three techniques together in Semi-ViT makes this comparison unfair to other methods.
The results with 100% data for Semi-ViT in Table 1 should be reported. No matter it's better, equal or worse than the baseline, it is valuable point to make fair comparison with the baselines (the MAE paper only reports the 100% data results). Similarly, 100% data results should be reported in Table 8.

---

> ### Author Response · Authors · 2022-08-01
> **Response to Reviewer Nfe1**
>
> Thanks for the valuable review! We provide the detailed responses to each question as below.
>
> **Q1: All the experimental results are as expected. I did not learn much new here.**
>
> A: To the best of our knowledge, ours is the first paper that shows that a pure ViT can achieve comparable or better results than a CNN for semi-supervised learning.  Specifically, we provide three novel insights: 1) Semi-ViT can achieve SOTA results in SSL, 2) the popular FixMatch is not stable for semi-supervised ViT, 3) probabilistic pseudo mixup can bring significant gains for SSL as an effective regularization.
>
> **Q2: Lack of novelty. The proposed three-stage training, EMA teacher, and the probabilistic Pseudo Mixup are all well-known techniques.**
>
> A: We are not claiming that the three-stage pipeline or the EMA-Teacher are a novel per se. What is novel is their use in achieving stable training in SSL, which is decisive to achieve state-of-the-art results with ViT. This stands in contrast with other choices, for instance SemiFormer’s use of FixMatch [60] which yields disappointing performance. The fact that our improvements are obtained using known techniques makes it simpler to understand and use, sparing others extensive experimentation to achieve stable training.
>
> What we do claim novelty for is the probabilistic pseudo mixup, which has shown nontrivial improvements over standard Pseudo Mixup, under different scenarios. It has some appealing properties that are not present in Pseudo Mixup. For example, although some samples don’t pass the confidence threshold, they still can contribute to the final loss, and their contributions depend on their confidence scores, so the low-confidence samples won’t hijack the training. As a result, our Semi-ViT is robust to confidence threshold, as also pointed out in the response to Q2 of Reviewer-mtWa.
>
> **Q3: Unfair comparison with SimCLRv2, PAWS, EMAN**
>
> A: We have described their pipelines and our differences with them in Section 2.1. SimCLRv2 also has a three-stage pipeline, but it uses knowledge distillation in their final stage. It would not be appropriate for us to change their approach for the purpose of comparison. In addition, the comparison with SimCLRv2 is fair, as the computations for the two pipelines are almost the same. Besides direct comparison of final results, our three-stage pipeline enables stable training for Semi-ViT and reduces the hyperparameter tuning, as seen in the extra ablation studies we provide in our response to other reviewers.
>
> **Q4: The results with 100% data for Semi-ViT**
>
> A: First, we would like to highlight that training a SSL method on 100% of the data is not a common procedure and not commonly reported in SSL papers. Instead, the community reports the performance of standard finetuning techniques on 100% of the data. Following the community, we report that upper bound in Tables 1 and 8.
>
> That said, we run the experiments that the reviewer requested (table below). Due to the limited time available, we did not have the opportunity to fully exploit hyperparameter tuning (i.e., results can get better), but still achieved encouraging preliminary result: about 0.5 point higher than finetuning on 100% of the data. Note that the improvement is not brought by longer training, because finetuning for 200 epochs won’t increase the accuracy. This experiment does show the robustness/generalization of our Semi-ViT.
>
> |          Method         | 1% | 10% | 100% |
> |:---------------------:|:----------------:|:----------------:|:----------------:|
> |         Finetune         |      57.4      |       73.1      |       83.7      |
> |       Semi-ViT-Base       |       71.0      |       79.7      |       84.2      |

---

> > ### Comment · Reviewer_Nfe1 · 2022-08-07
> > **thank authors for reply, but keep my rating**
> >
> > Thank authors for replying the review! However, the rebuttal does not change my understanding of the work and its results.

---

> > > ### Author Response · Authors · 2022-08-07
> > > **follow-up response**
> > >
> > > Dear reviewer, thanks for taking your time to read our responses. We have tried our best to answer your questions and address your concerns. Is there still any further confusion or concern we can help you to address? If it is still about the technical novelty, we appreciate if the reviewer could also read the other reviews to have a comprehensive evaluation. The code will also be provided to reproduce the results.

---

### Meta-Review · Area_Chair_rLCC · 2022-08-26

**Recommendation:** Accept
**Confidence:** Certain

**Metareview:**

This paper explores Semi-ViT, a semi-supervised learning approach for vision transformers. Semi-VIT build-on three stages pipeline such as SimCLRv2. The authors introduce a probabilistic mixup for the semi-supervised finetuning stage which gives consistent experimental improvements.  Semi-ViT shows strong empirical results as it achieves 80% top-1 accuracy on ImageNet using only 1% labels, which is comparable with Inception-v4 using 100% ImageNet labels,


Demonstrating that ViT+semi-supervised training enables to reach 80% top-1 accuracy on 1% ImageNet is novel and of potential interest to the SSL community.  I therefore recommend acceptance. However, I would encourage the authors to clarify that the three-stages pipeline is not a contribution of the paper and focus the novelty on the probabilistic mixup and the experimental study.

**Award:**

No

---

### Decision · Program_Chairs · 2022-09-14

Accept